# High frequency acoustic cell stimulation promotes exosome generation regulated by a calcium-dependent mechanism

Lizebona August Ambattu [ID] [1], Shwathy Ramesan[1], Chaitali Dekiwadia[2], Eric Hanssen [ID] [3], Haiyan Li[4] & Leslie Y. Yeo [ID] [1✉]

Exosomes are promising disease diagnostic markers and drug delivery vehicles, although their use in practice is limited by insufficient homogeneous quantities that can be produced. We reveal that exposing cells to high frequency acoustic irradiation stimulates their generation without detriment to cell viability by exploiting their innate membrane repair mechanism, wherein the enhanced recruitment of calcium ions from the extracellular milieu into the cells triggers an ESCRT pathway known to orchestrate exosomal production. Given the high post-irradiation cell viabilities (≈95%), we are able to recycle the cells through iterative irradiation and post-excitation incubation steps, which facilitate high throughput production of a homogeneous population of exosomes—a particular challenge for translating exosome therapy into clinical practice. In particular, we show that approximately eight- to ten-fold enrichment in the number of exosomes produced can be achieved with just 7 cycles over 280 mins, equivalent to a yield of around 1.7–2.1-fold/h.

---

[1] Micro/Nanophysics Research Laboratory, School of Engineering, RMIT University, Melbourne, VIC 3000, Australia. [2] RMIT Microscopy and Microanalysis Facility, RMIT University, Melbourne, VIC 3000, Australia. [3] Advanced Microscopy Facility, Bio21 Molecular Science & Biotechnology Institute, The University of Melbourne, Parkville, VIC 3010, Australia. [4] School of Biomedical Engineering & Med-X Research Institute, Shanghai Jiao Tong University, Shanghai 200030, China. ✉email: leslie.yeo@rmit.edu.au

Exosomes are extracellular vesicles (EVs) between 30 and 150 nm in diameter that are secreted by all eukaryotic cells into the extracellular microenvironment. Unlike other subclasses of EVs, they are released from multivesicular bodies rather than directly from the plasma membrane. Exosomes play an important role in intercellular communication by facilitating the transmission of macromolecules such as mRNA, miRNA, DNA, lipids and proteins between cells, and therefore their influence have been implicated in disease development and transmission[1–3]. As such, there has been compelling interest for their isolation from circulatory samples for disease detection, particularly cancer[4–10]. Exosome-based liquid biopsies, for example, allow real-time profiling of a patient's tumour activity by isolating DNA or RNA from the exosome for further analysis, thus facilitating monitoring of disease progression without requiring invasive surgical procedures[11,12].

Being the facsimile of the cell, exosomes are also superior drug delivery vectors compared to synthetic polymers or viruses since their lipid bilayer structure is similar to that of cell membranes and are therefore considerably less likely to invoke an immune response[13–18]. Besides their ability to traverse the blood–brain barrier[19–24], these azoic entities are, in addition, known to induce transcriptomic and phenotypic changes[25,26] and therefore play critical roles in stem cell differentiation and modulating the tumour niche[27–29]. Moreover, as all eukaryotic cells produce exosomes and internalise them, they are able to target any cell type and have recently been used for therapeutic targeting of the oncogene KRAS, considered among the most challenging of drug targets[30]. Consequently, there are currently widespread efforts to harness them as carriers in gene and protein therapy.

A critical technical challenge in practice, however, lies in the crucial need to obtain adequate quantities of pure exosomes that are sufficiently homogeneous through cell culture (over several days) and subsequently isolating them[30,31]. A number of methods to enhance exosome yield have therefore been proposed[32]. These, for example, involve chemical (e.g. ionomycin[33] or intracellular calcium[34,35]), biochemical (e.g. extracellular DNA[36] or liposomes[37], or altering proteomic content such as through the introduction of p53[38]), pH[39] or mechanical (e.g. cyclic stretching)[40] stimuli; methods to induce cell hypoxia[41]; cytoskeletal protein alteration[42]; gene overexpression[43] or exposure to thermal, oxidative, photodynamic or radiative stress[44–46]. More recently, a cell nanoporation technique that can be scaled for high throughput has also been demonstrated[47].

There are nevertheless a number of potential disadvantages to some of the aforementioned methods. The use of additives such as ionomycin and calcium phosphate, for example, while capable of enhancing exosome yield by 2.5-fold within 2–72 h (approximately 0.03–1-fold/h), are, however, dose dependent and overexposure of the cells to these chemicals can lead to a considerable reduction in their viability[34]; similarly, exposing cells to ionising radiation can result in cell apoptosis[46]. Thermal and oxidative stresses, on the other hand, have been reported to increase exosome yield by approximately 20–30-fold in 24 h (approximately 0.8–1.25-fold/h) but can generate immunoresponsive exosomes, which could impair their diagnostic or therapeutic potential[44]. Given the role of heat-shock responses in the exosome production mechanism in the cell nanoporation technique[47,48], a corollary to the enhancement afforded by the technique is the upregulation in p53 tumour-suppressor protein activity, which can potentially result in undesirable development of a proinvasive microenvironment[38,49]. In any case, besides addressing low exosome yield, few of these methods, if any, are also able to circumvent lipidome and proteome heterogeneity in the exosome population, which constitutes a further barrier to translation of exosome therapies into clinical practice[50,51].

We show in this work that it is possible to obtain a 1.7-fold increase in exosome production in mammalian cells (U87-MG and A549 cells; cancer cell lines were chosen to demonstrate proof of concept as they are commonly used in exosome release studies, and since exosomes derived from cancer cells have been identified as potential cancer biomarkers[4] and vaccine candidates[52]) by exposing them to low power (approximately 4 W) MHz-order acoustic irradiation in the form of surface-reflected bulk waves (SRBWs)—high-frequency (10 MHz order) electromechanical hybrid surface and bulk waves that propagate through a piezoelectric substrate[53]—for just 10 min followed by a short 30-min incubation period. In addition to showing that a very high proportion (approximately 95%) of the irradiated cells remained adherent and viable and continued to proliferate normally, we elucidate the possible pathway through which the exosome production and secretion are enhanced by the acoustic stimulation. Given such high post-excitation cell viabilities, a unique advantage of the technology is the opportunity for the same cell source to be repeatedly cycled through successive irradiation and post-excitation incubation steps, thus allowing a means not only for increasing production throughput but also avoiding proteome and lipidome heterogeneity in the exosome population, which is problematic for exosome therapeutics. As a demonstrative example, we show the possibility of achieving an 8–10-fold increase in exosome production with 7 cycles over 280 min, equivalent to an exosome production yield of approximately 1.7–2.1-fold/h.

## Results

**High-frequency acoustic stimuli enhances exosome production.** Figure 1b shows the acetylcholine esterase activity that quantifies the presence of U87-MG exosomes in the sample as a function of the post-excitation incubation period following 10 min of SRBW excitation using the set-up shown in Fig. 1a and Supplementary Fig. 1; parenthetically, we note that although the esterase activity, in general, is not just specific to exosomes but rather to all EVs, these measurements were carried out after the exosomes were purified from the EVs in the spent media using an exosome isolation kit and hence the esterase activity in this case can be considered as a positive marker for the exosomes. It can be seen that the esterase activity increased markedly by approximately 1.7-fold in the first 30 min following application of the SRBW irradiation for 10 min to the cells, suggesting elevated levels of exosomes that were secreted by the cells within this period, after which the number of exosomes gradually reduced with increasing incubation time, possibly due to their internalisation by neighbouring cells[54,55], as evident from the images in Supplementary Fig. 2 that show their uptake over 18 h. Increasing the input power to the device and hence the SRBW excitation amplitude, on the other hand, can be seen in Fig. 1c to increase exosome production, although we note that increasing the power beyond approximately 4 W (as well as the exposure time beyond 10 min) leads to a reduction in the cell viability (Supplementary Fig. 3a, b). As such, the post-excitation incubation period was therefore fixed at 30 min and the input power to the device at 4 W in all subsequent experiments to maximise both exosome production and cell viability.

Figure 1d, e (see n = 1 data) shows that the majority of U87-MG cells (approximately 95%) remained viable and adherent following SRBW irradiation and continued to proliferate normally, consistent with results from preceding studies employing similar high-frequency acoustic forcing for intracellular macromolecular uptake[56]. Similar results were also observed for A549 cells (see Supplementary Fig. 4b, c). Unlike low-frequency ultrasound (10–100 kHz up to 1 MHz) typically used in sonoporation, the considerably higher frequencies and lower powers (one to two

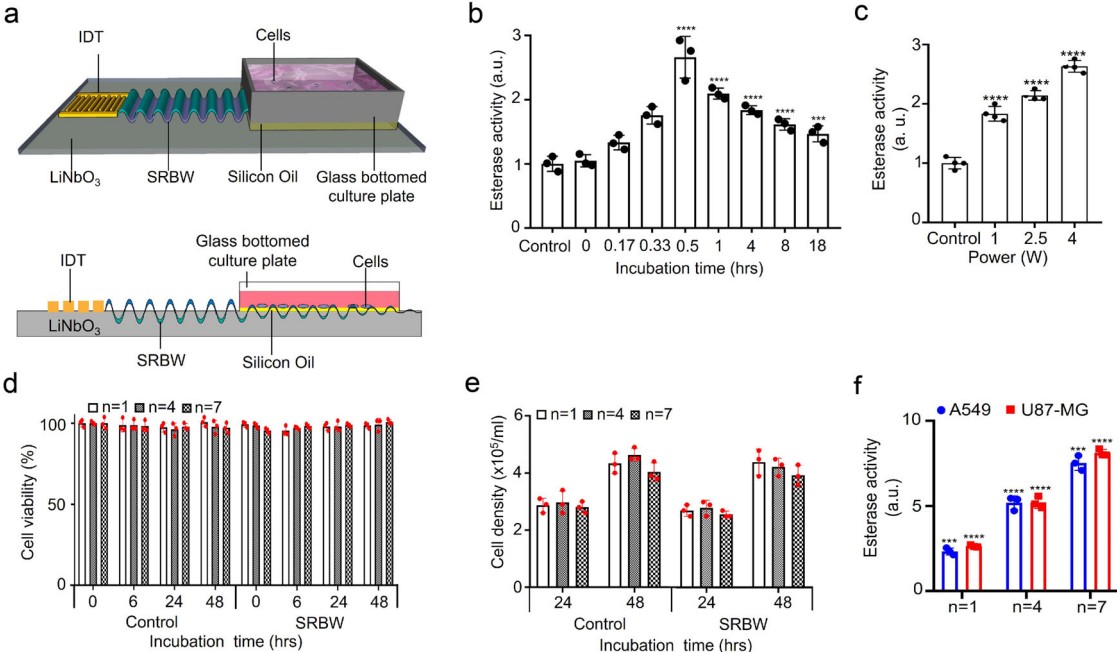

**Fig. 1 Experimental set-up, EV acetylcholine activity and parent cell viability. a** Perspective and side view schematics (see also images in Supplementary Fig. 1) of the experimental set-up in which the SRBW (not to scale), generated along a piezoelectric lithium niobate ($LiNbO_3$) substrate by applying an AC electric signal at the device's resonant frequency (10 MHz) to an interdigitated transducer electrode (IDT) photolithographically patterned on the substrate, is coupled through a thin layer of silicon oil into a glass-bottom culture plate containing the adherent cells to stimulate their production of exosomes. **b** Acetylcholine esterase activity of EVs isolated from spent U87-MG cell media as a function of the post-excitation (i.e., after 10 min of SRBW exposure) incubation time relative to that for the control sample, which comprised unexposed cells incubated over the same period. **c** Enhancement in U87-MG EV production under SRBW excitation with increasing input power to the device, as indicated by the increase in esterase activity in the spent cell media relative to the unexposed control; the post-excitation incubation period in all cases was fixed at 30 min. Cell viability data at higher powers beyond 4 W are shown in Supplementary Fig. 3a. **d** U87-MG cell viability, as measured from an MTT assay, **e** U87-MG cell population density and **f** relative esterase activity of U87-MG (red squares) and A549 (blue circles) EVs following successive 10-min excitation and 30-min incubation cycles $n$ ($n = 1$: no shading, $n = 4$: fine shading, $n = 7$: coarse shading). The data are represented in terms of the mean value ± the standard error over triplicate runs, and the asterisks *** and **** indicate statistically significant differences with $p < 0.001$ and $p < 0.0001$, respectively. The corresponding results for A549 cells can be found in Supplementary Figs. 3 and 4, although we have included the A549 cell esterase activity data for successive cycles in **f**.

orders of magnitude) associated with the SRBW excitation or its surface acoustic wave counterpart do not generate any appreciable cavitation[57] to induce pore formation in the cell plasma membrane, which is known to inflict considerable damage to the cell. Rather, it was postulated in ref. [56] that the high-frequency excitation was only sufficient to drive reversible permeabilisation of the membrane by inducing transient structural reorganisation of the lipids that make up the plasma membrane[56,58], which immediately reseals upon relaxation of the acoustic signal. This would not only explain the high viabilities observed in the present work but also suggests the possibility that the acoustic excitation could also be responsible for enhancing the secretion of the exosomes produced under the same stimuli.

Moreover, the high cell viability offers the unique possibility for further increasing the exosome yield from the same cell population by repeatedly exposing the same batch of cells to successive excitation–incubation cycles, each cycle $n$ comprising SRBW irradiation for 10 min followed by a 30-min incubation period. As shown in Fig. 1f, fourfold and eightfold increases in the relative esterase activity after $n = 4$ and $n = 7$ cycles, corresponding to a total duration of 160 and 280 min, respectively, were observed without any appreciable effects on the cell homoeostasis, i.e. no substantial decreases could be seen in the viability of the cells (Fig. 1d and Supplementary Fig. 4b) or their ability to proliferate (Fig. 1e and Supplementary Fig. 4c) compared to the untreated cells over the same period. We note the possibility of

some exosomes being trapped in the membranes during their isolation[56], and hence the likelihood that the number of exosomes produced could be higher in each successive cycle. Moreover, given that the cell viabilities are maintained even after 7 cycles, it is possible to continue the excitation–incubation cycles to further increase the exosome yield—such an ability to recycle the cells constitutes a significant advantage over other methods, both in terms of maintaining proteome and lipidome homogeneity in the exosome population, which is highly desirable and a significant challenge at present for exosome therapeutics[50,51], and in reducing the cost of the cell feedstock, which can be considerable, particularly for large-scale exosome manufacture[59].

The number concentration and size distribution of the exosomes isolated from the control and irradiated cells after $n = 7$ cycles, obtained via nanoparticle tracking analysis (NTA) and dynamic light scattering (DLS), respectively, are shown in Fig. 2a, b, respectively (the size distribution for exosomes isolated from A549 cells can be found in Supplementary Fig. 4a). While the concentration verifies the enrichment (approximately tenfold; cf. eightfold enrichment obtained through quantification with the esterase activity in Fig. 1f) in the exosome production obtained from the iterative excitation–incubation steps, a comparison of the sizes of the exosomes between the control and irradiated samples not only shows slightly smaller hydrodynamic diameters but also indicates that a large proportion of the EVs that were produced under SRBW irradiation appear to consist primarily of

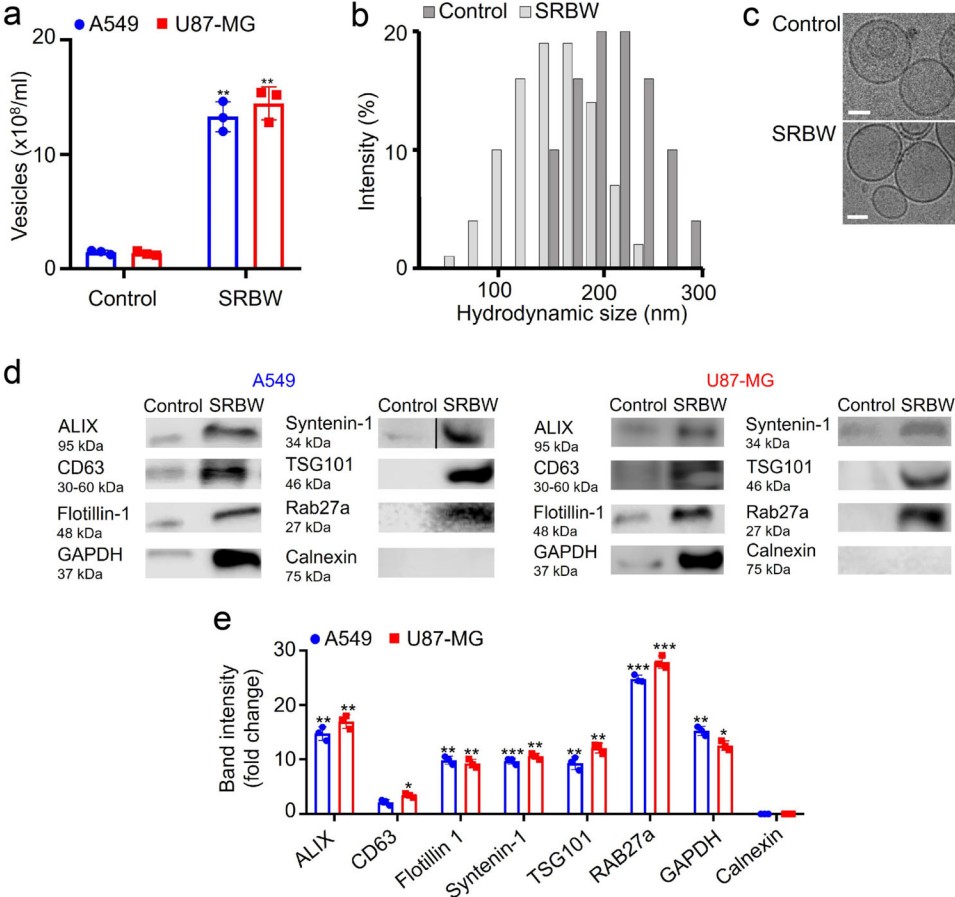

**Fig. 2 Exosome characterisation.** Comparison of **a** the number concentration, obtained through NTA, **b** the hydrodynamic size distribution, obtained through DLS, **c** representative cryo-EM images, **d** the protein profile, obtained via western blotting, and **e** the band intensities of EVs isolated from the unexposed control (dark grey bars in **b**) and SRBW-treated (light grey bars in **b**) U87-MG cells (aside from **a**, **e** where blue circles indicate EVs isolated from A549 cells and red squares indicate EVs from U87-MG cells) after 7 successive excitation–incubation cycles; the scale bars in **c** represent a length of 50 nm. The data are represented in terms of the mean value ± the standard error over triplicate runs, and the asterisks *, ** and *** indicate statistically significant differences with $p < 0.05$, $p < 0.01$ and $p < 0.001$, respectively. We note that the blot image for syntenin-1 has been spliced, as demarcated with a dividing line; a full scan of the entire blot is provided in Supplementary Fig. 5.

exosomes, which, by definition, have size ranges between 30 and 150 nm. Taken together with the sphericity of the entities observed in the cryo-electron microscopic (cryo-EM) images in Fig. 2c, which are representative across all of the results obtained (more representative images can be found in Supplementary Fig. 6a), this suggests minimal, if not negligible, formation of apoptotic bodies, which are usually irregular in dimension. Moreover, we note from the cryo-EM images that the membrane integrity of the exosomes obtained following SRBW irradiation appears to be preserved.

**Mechanism for exosome production.** As physical characterisation does not completely rule out the existence of most other classes of EVs such as microvesicles and apoptotic bodies, we look to evidence beyond physical characterisation, in particular exosomal protein profiling via western blotting, which may allude to the mechanism by which exosomes are generated, and, in doing so, verify the existence of the exosomes produced through acoustic stimulation. Figure 2d, e reveals an abundance of proteins in the exosome lysate after $n = 7$ successive excitation–incubation cycles that are implicated in exosome biogenesis following SRBW exposure, specifically those involved in the ESCRT (endosomal sorting complexes required for transport) pathway that orchestrates the generation of

late endosomes, i.e. multivesicle bodies (MVBs) within the cell, whose fusion with the plasma membrane leads to the release of intraluminal vesicles into the extracellular matrix as exosomes (the transmission electron microscopy (TEM) images of fixed cells following acoustic excitation in Supplementary Fig. 6b shows an increase in MVBs within the cells compared to the control). In particular, we note the overexpression of ALIX (ALG-2 (apoptosis-linked gene 2)-interacting protein X) and TSG101 (tumour-suppressor gene 101)—the two accessory proteins involved in the ESCRT machinery—and CD63, which has been reported to also be present in the ESCRT pathway, in addition to that of other exosomal markers, namely, flotillin-1, essential for membrane invagination that is a precursor to MVB formation, syntenin-1, a cargo sorting protein without which exosomes cannot be generated, and Rab27a, which facilitates MVB docking onto the plasma membrane and whose elevated levels do not only imply an enhancement of exosome production in the cell but also an increase in their secretion from the cell[60–65]. Correspondingly, we note the absence of calnexin—an endoplasmic reticulum (ER) marker that constitutes a negative control in exosome production[66]—that also confirms the absence of microvesicles[67] and apoptotic bodies in the irradiated sample[68,69].

Moreover, the exosomal protein expression can also be seen to increase with the number of excitation–incubation cycles (Fig. 3). In particular, we observe the ALIX (Fig. 3b) and syntenin-1

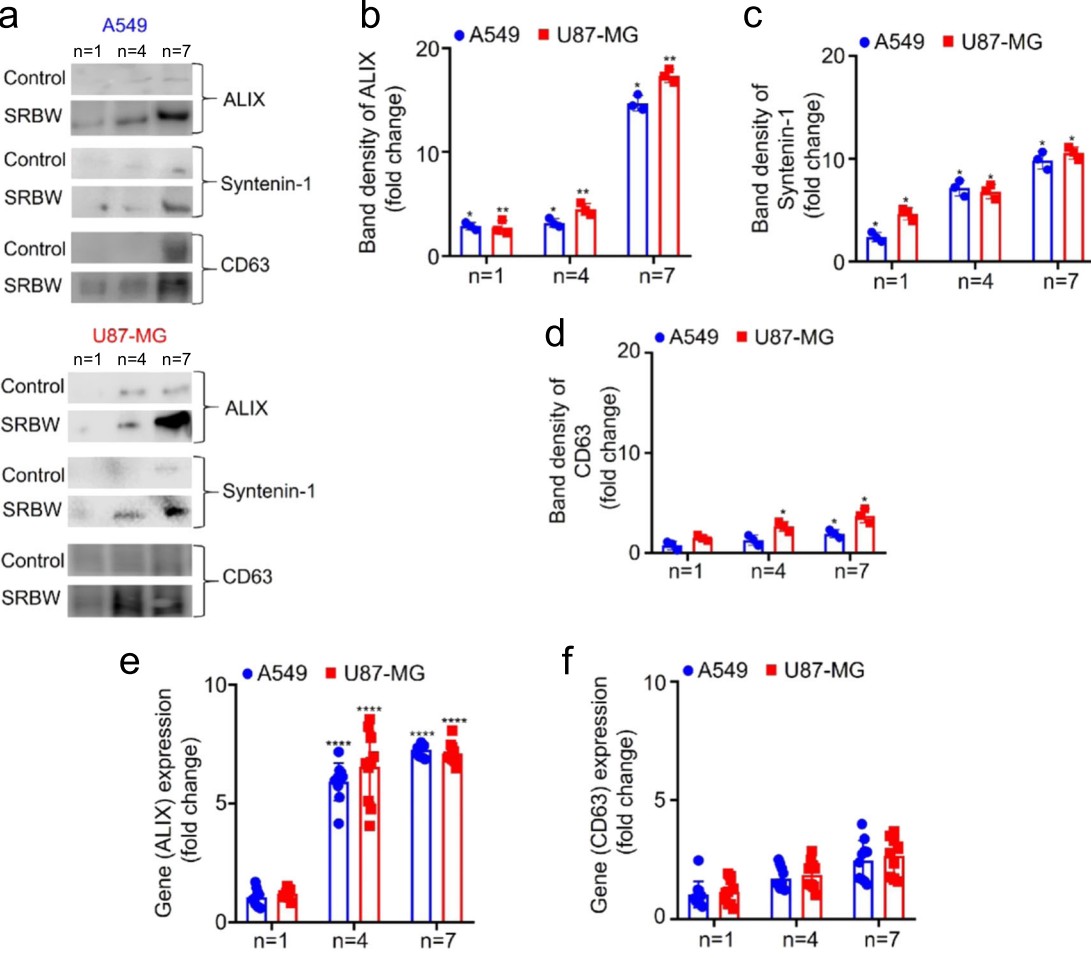

**Fig. 3 Exosome protein profiling.** Exosome protein profiling via **a** western blot analysis, showing the progressive increase in the relative band intensity of **b** ALIX, **c** syntenin-1 and **d** CD63 with successive number of excitation–incubation cycles $n$, compared to the unexposed control over the same duration. **e, f** RT-qPCR analysis quantifying the mRNA expression of ALIX and CD63 between the unexposed control and SRBW-irradiated A549 (blue circles) and U87-MG (red squares) cells after successive number of cycles, normalised against GAPDH. The data are represented in terms of the mean value ± the standard error over triplicate runs (10 runs for the RT-qPCR experiments), and the asterisks *, ** and **** indicate statistically significant differences with $p < 0.05$, $p < 0.01$ and $p < 0.0001$, respectively.

(Fig. 3c) levels to increase progressively with successive cycling, whereas the CD63 (Fig. 3d) level can be seen to increase initially before plateauing between $n = 4$ and $n = 7$, consistent with the mRNA expression levels in Fig. 3e, f, measured using real-time quantitative reverse transcription polymerase chain reaction (RT-qPCR) with glyceraldehyde 3-phosphate dehydrogenase (GAPDH) as a housekeeping gene. This suggests that the SRBW-irradiated cells might favour an ALIX-mediated pathway in which the ESCRT accessory component ALIX complexes with syntenin-1 to regulate its role in membrane recruitment and intraluminal budding[70]; the relatively lower increase (approximately threefold) in CD63 expression compared to that for ALIX (approximately sixfold) is likely because the affinity of CD63 to syntenin-1 is roughly ten times less than that of ALIX[62].

The role of the acoustic stimuli in the overexpression of exosomal proteins can be understood from intracellular calcium profiling since intracellular calcium plays a crucial role in ESCRT recruitment and hence endosomal release. More specifically, it has been reported that cells under stress are typically associated with increased calcium ion ($Ca^{2+}$) levels, either due to its release from the intracellular $Ca^{2+}$ store or through its uptake into the cell from the extracellular milieu[33,34,71]. That the acoustic stimulation increases intracellular $Ca^{2+}$ through the latter mechanism, i.e. internalisation of $Ca^{2+}$ from the extracellular milieu, is evident

from the measurements of the intracellular $Ca^{2+}$ level in Fig. 4a, which shows an elevated reading (iii) for the SRBW irradiated sample above the baseline level associated with its unexposed counterpart (i). This is further verified in the case when no $Ca^{2+}$ was present in the extracellular milieu (ii), in which case no substantial change in the intracellular $Ca^{2+}$ level compared to the control (i) was observed even when the cells were exposed to the acoustic irradiation.

A similar increase in the intracellular $Ca^{2+}$ level for the acoustically irradiated cells (vii) can be seen even in the presence of calcium channel blockers amiloride HCl and thapsigargin and a membrane permeable intracellular $Ca^{2+}$ chelator bis-acrylamide, 1,2-bis(2-aminophenoxy)ethane-$N,N,N',N'$-tetraacetic acid (BAPTA-AM), which act to deplete the intracellular calcium store (iv, v, vi). We note that such an increase in the intracellular $Ca^{2+}$ was also observed when cells were exposed to similar high-frequency vibrational excitation in ref. [56] and is likely due to the increase in membrane permeability as a consequence of the transient reorganisation of the plasma membrane lipid structure during high-frequency acoustic stimulation. Similar trends can be seen in the acetylcholine esterase activity (Fig. 4b) and the mRNA overexpression associated with ALIX and CD63 (Fig. 4c) wherein we observe across-the-board enhancement in exosome production with SRBW irradiation even in the presence of the

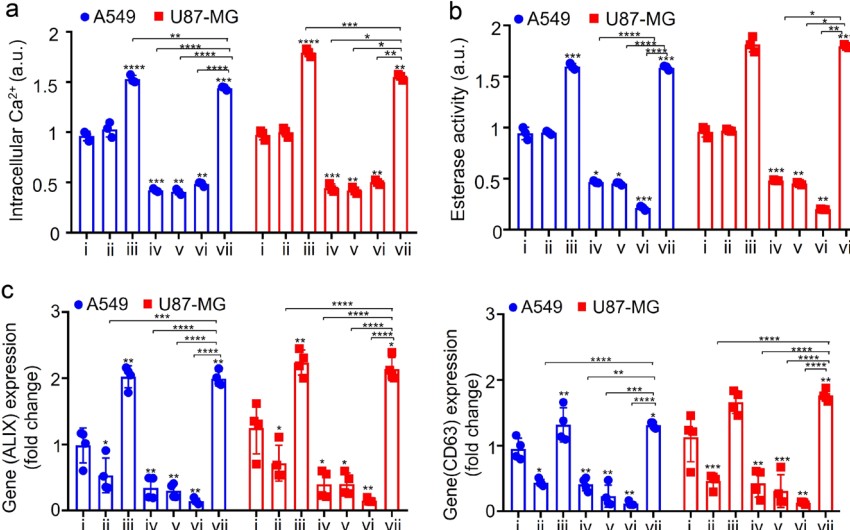

**Fig. 4 Intracellular calcium profiling. a** Intracellular $Ca^{2+}$ levels, measured from a Fura-2 AM assay, **b** acetylcholine esterase activity and, **c** mRNA expression of ALIX and CD63, as quantified via RT-qPCR analysis, for A549 (blue circles) and U87-MG (red squares) cells exposed to the SRBW irradiation compared to the unexposed control (i), in the absence (−) and presence (+) of extracellular calcium as well as a combination of a $Ca^{2+}$ inhibitor (thapsigargin), ion channel blocker (amiloride HCl) and membrane-permeable intracellular $Ca^{2+}$ chelator (BAPTA-AM); see the key in Table 1. The data are represented in terms of the mean value ± the standard error over quadruplicate runs, and the asterisks *, **, *** and **** indicate statistically significant differences with $p < 0.05$, $p < 0.01$, $p < 0.001$ and $p < 0.0001$, respectively.

| Table 1 Key to Fig. 4. | | | | | | | |
|---|---|---|---|---|---|---|---|
| | **i** | **ii** | **iii** | **iv** | **v** | **vi** | **vii** |
| Amiloride HCL | − | − | − | + | + | + | + |
| BAPTA-AM | − | − | − | + | + | + | + |
| Thapsigargin | − | − | − | + | + | + | + |
| Extracellular calcium | + | − | + | − | + | − | + |
| SRBW | − | + | + | + | − | − | + |

inhibitor(s) and/or chelator (the difference in the relative levels between the ALIX and CD63 expression levels in Fig. 4c with those in Fig. 3e, f arises because ALIX alone is calcium dependent; as such, only the expression levels of ALIX and not CD63 is expected to change appreciably in the presence of the $Ca^{2+}$ inhibitors/chelators[72,73]), thus highlighting the essential role of intracellular $Ca^{2+}$ in producing transcriptomic changes under the SRBW stimuli.

Taken together, these results suggest that the increase in intracellular $Ca^{2+}$ uptake into the cell in response to high-frequency stimulation has a twofold effect. In addition to directly enhancing intracellular transport across the plasma membrane due to its permeabilisation as a consequence of the acoustically driven vibrational stressing of the membrane[56], the immediate healing of the membrane upon relaxation of the SRBW excitation, given the transient and reversible nature of this process involving rearrangement of its lipid structure, also leads to recruitment of extracellular $Ca^{2+}$. This then prompts recruitment of ESCRT nucleating factors ALIX and TSG 101, which form a complex with syntenin-1 at the site of repair, consistent with previous studies where ALIX- and CD63-positive vesicle release was observed in response to $Ca^{2+}$-triggering following membrane puncture[71,74,75]. The requirement for $Ca^{2+}$ to be present in the extracellular environment for its internalisation into the cell to trigger ESCRT recruitment and endosomal release suggests that the SRBW irradiation does not induce ER stress to trigger the release of $Ca^{2+}$ from the internal cellular store. That the high-frequency acoustic irradiation is capable of altering cellular activity without imparting

ER stress, as confirmed by the absence of calnexin—an ER stress marker—in the protein profile of the isolated EVs in Fig. 2e, is unique and quite unlike other techniques for enhancing exosomal yield that involve application of external stimuli to the cell.

## Conclusions

We show that low-level insults involving high-frequency acoustic stimulation to mammalian cells enhances production of EVs through a calcium-dependent ALIX-mediated pathway while maintaining very high (≈95%) cell viability. In addition to showing that the EVs that are generated primarily consist of exosomes, we elucidate via protein and calcium profiling the mechanism by which such enhancement in exosome production transpires. In particular, the gentle vibration of the cells at high frequencies drives transient reorganisation of the lipid structure of the plasma membrane that increases its permeability without inflicting significant damage (e.g. via poration) to it. This augmentation in membrane permeability, together with its healing when the acoustic signal is relaxed, promotes recruitment of calcium ions into the cell, initiating the assembly of ESCRT accessory proteins at the site of repair, which, in turn, orchestrates the cascade of events—MVB fusion, intraluminal vesicle accumulation and cargo release—that lead to the production of exosomes.

Through an iterative procedure in which the cells are repeatedly exposed to cycles of acoustic irradiation for 10 min followed by 30 min of post-excitation incubation, we show that it is possible to obtain a 8–10-fold amplification in the number of exosomes in just 7 cycles corresponding to a total treatment duration of 280 min, which is equivalent to an approximate yield of 1.7–2.1-fold/h. This scalable (through massive parallelisation, given the low cost of the SRBW devices (around US$1/device), achieved by exploiting the economies of scale through mass nanofabrication—see, for example, the discussion in ref. [76]) platform thus offers a facile means by which the current bottlenecks in exosome technology (namely, the inability to adequately produce the large amounts required that are sufficiently homogeneous from the same cell source for clinical use) can be circumvented, therefore offering a potential solution that enables the

exciting promise of exosomes for diagnostics and therapeutics to be realised.

## Methods

**Materials.** Unless otherwise specified, sodium chloride, methanol, ethanol, iso-propanol, liquid ethane, RNase-free water, nuclease-free water, glycerol, glycerine, non-fat skimmed milk, silicon oil, dimethylsulfoxide (DMSO), sodium cacodylate buffer, uranylacetate, $\beta$-mercaptoethanol, Tween® 20, sodium dodecyl sulfate (SDS), Trizma® (Tris) base, phosphate-buffered saline (PBS), amiloride hydro-chloride (HCl), Tris–HCl, chloroform, ammonium persulfate, BAPTA-AM, Gibco penicillin–streptomycin, acetylthiocholine, 5,5′ dithiobis(2-nitrobenzoic acid), ethylenediaminetetraacetic acid (EDTA), trypsin–EDTA, paraformaldehyde, glu-taraldehyde, osmium tetroxide, potassium ferrocyanide, Triton™ X-100, thapsi-gargin, bovine serum albumin, foetal bovine serum (FBS), Roswell Park Memorial Institute (RPMI) 1640 medium, Dulbecco's Modified Eagle Medium (DMEM) without calcium, Dulbecco's PBS, bromophenolblue, radioimmunoprecipitation (RIPA) assay buffer, biotinylated protein ladder, Hoechst 33342, Trypan Blue solution, Fura-2 acetoxymethyl ester (AM), 3-(4,5-dimethylthiazol-2-yl)-2,5-diphenyltetrazolium bromide (MTT), BODIPY™ TR ceramide, Vybrant® MTT Cell Proliferation Assay Kit, Bicinchoninic Acid (BCA) Protein Assay Kit, Pure-Exo® Exosome Isolation Kit (101 Bio LLC, Mountain View, CA, USA), Luna-Script® RT SuperMix Kit, Luna® Universal qPCR Master Mix, TRiZOL™ reagent, Pierce™ ECL Western blotting detection reagent, Precision Plus Protein™ WesternC™ Standards (Bio-Rad Laboratories, Gladesville, NSW, Australia), Bio-tinylated Protein Ladder Detection Pack (Cell Signaling Technology Inc., Danvers, MA, USA), nitrocellulose membrane (0.45 μm), protease inhibitor cocktail tablet, polyacrylamide gel, Formvar/carbon-coated and holey carbon grids (Emgrid Pty. Ltd., Gulfview Heights, SA, Australia), A549 and U87-MG cells (American Type Culture Collection, Manassas, VA, USA), Exosome Spin Columns (MW 3000), T25 cell culture flask, MatTek 24-well glass-bottom plates and Nunc™ Lab-Tek™ II Chamber Slide and Chambered Coverglass were acquired from Thermo Fischer Scientific Pty. Ltd. (Scoresby, VIC, Australia).

Anti-GAPDH mouse antibody, anti-ALIX mouse antibody, anti-Rab 27a rabbit antibody, anti-mouse horse radish peroxidase (HRP)-conjugated antibody, anti-biotin HRP-linked antibody and anti-rabbit HRP-conjugated antibody were obtained from Cell Signaling Technology Inc. (Danvers, MA, USA), anti-TSG101 mouse antibody and anti-syntenin-1 rabbit antibody from Thermo Fisher Scientific Pty. Ltd. (Scoresby, VIC, Australia), anti-calnexin rabbit antibody from Abcam (Cambridge, UK), anti-flotillin-1 mouse antibody from BD Biosciences (San Jose, CA, USA) and anti-CD63 mouse antibody from Invitrogen (Carlsbad, CA, USA).

The following primers used for the RT-qPCR analysis were acquired from Integrated DNA Technologies Inc. (Coralville, IA, USA):

ALIX (forward): 5′-GACGCTCCTGAGATATTATGATCAGA-3′,
ALIX (reverse): 5′-ACACACAGCTCTTTTCATATCCTAAGC-3′,
CD63 (forward): 5′-TAGATTCGGCAGCCATGGCGGTGGAA-3′,
CD63 (reverse): 5′-ACTGACCAGACCCCTACATCACC-3′,
GAPDH (forward): 5′-CATGTTCCAATATGATTCCACC-3′,
GAPDH (reverse): 5′-GATGGGATTTCCATTGATGAC-3′.

**Device fabrication.** The SRBW devices, schematically illustrated in Fig. 1a and shown in the images in Supplementary Fig. 1, comprised 500-μm-thick 127.86° $Y$–$X$ rotated lithium niobate (LiNbO$_3$) single-crystal piezoelectric substrates (Roditi Ltd., London, UK) on which 40 alternating finger pairs of 11-mm-wide and 66-nm-thick straight aluminium interdigitated transducer (IDT) electrodes in a basic full-width interleaved configuration were patterned atop a 33-nm-thick chromium adhesion layer through sputter deposition and standard ultraviolet (UV) photolithography. The width and the gap of the IDT fingers ($\lambda/4$) then sets the SRBW wavelength $\lambda = 398$ μm and hence the device's resonant frequency $f = 10$ MHz. To generate the SRBW, an alternating electrical signal is applied to the IDTs at the resonant fre-quency using a signal generator (SML01, Rhode & Schwarz Pty. Ltd., North Ryde, NSW, Australia) and amplifier (10W1000C, Amplifier Research, Souderton, PA, USA). As depicted in Fig. 1a, a thin layer of silicon oil with viscosity 45–55 cP and density 0.963 g/ml at 25 °C was sandwiched between the SRBW device and the glass-bottom chamber slide in which the cells were contained to aid coupling of the acoustic energy from the device into the wells.

**Cell culture and acoustic exposure.** U87-MG human glioblastoma cells and A549 adenocarcinomic human alveolar basal epithelial cells were, respectively, cultured in DMEM and RPMI medium supplemented with 10% FBS and 1% penicillin–streptomycin (100 units/ml) in a humidified incubator maintained at 37 °C and 5% CO$_2$. The cells were grown in a standard T25 flask until they reached 80–90% confluency, following which they were detached using 0.05% trypsin–EDTA 24 h prior to the experiments, reseeded in the 8-well plates at a density of 300,000 cells per well and incubated for 18 h. The cells were then thrice washed with PBS and replenished with exosome-depleted medium (DMEM with 1% penicillin–streptomycin and 10% exosome-depleted FBS, the latter prepared by centrifuging at 121,800 × g for 19 h from which the supernatant was filtered using a 0.22-μm filter and used immediately or stored) for 48 h; this washing and replenishing step was also repeated immediately prior to the experiment. The cells in the well plate were then irradiated with the SRBW at the prescribed input power to the device for the stipulated duration. Following cessation of the acoustic field, the cells and media were incubated at 37 °C for the prescribed time period, following which the spent culture media was immediately collected and the exosomes isolated for further characterisation. For the control, the cells were seeded at the same density and incubated over the same time period.

**Cell viability and proliferation.** The viability of cells following their exposure to the SRBW irradiation was assessed using an MTT proliferation assay in which the treated cells were washed with PBS immediately after collecting the spent media following which MTT solution at a final concentration of 0.5 mg/ml in serum-free medium was added to each well and incubated for 3 h. The absorption of formazan crystals dissolved in DMSO was measured at 570 nm using a spectrophotometric plate reader (CLARIOstar®, BMG LabTech, Mornington, VIC, Australia) and normalised with respect to the absorbance of the control containing cells at the same concentration that were not exposed to the SRBW irradiation but incubated for the same time period. The viability of the SRBW-treated cells after 6, 24 and 48 h was also analysed to determine long-term cytotoxicity effects. The cells' ability to continue to proliferate following exposure to the SRBW irradiation, on the other hand, was evaluated using a Trypan Blue exclusion assay in which the SRBW-treated cells were trypsinised immediately and reseeded. The cell count was then determined after 24 and 48 h using Trypan Blue (0.4%) solution with a cell counter (Invitrogen Countess™, Thermo Fisher Scientific Pty. Ltd., Scoresby, VIC, Australia).

**Exosome isolation, quantification and characterisation.** The spent medium, collected from the SRBW-treated and control (untreated) samples following the stipulated excitation and subsequent incubation period, was centrifuged at 2000 × g for 15 min at 4 °C, from which the supernatant was collected and filtered using the PureExo® Exosome Isolation Kit. The isolated exosomes were stored at 4 °C for a week or at −80 °C for up to 3 months. Total exosomal protein content was estimated using BCA analysis in which 5 μl of the exosome isolate was recon-stituted in PBS to 150 μl and mixed with BCA reagent at a 1:1 volume ratio prior to incubation at 37 °C for 2 h, after which the solution was brought to room tem-perature and its absorbance was measured at 562 nm with a spectrophotometric plate reader (CLARIOstar®, BMG LabTech, Mornington, VIC, Australia).

The exosomes released were quantified by measuring their acetylcholine esterase activity. Briefly, 25 μl of the exosome isolate was suspended in PBS (pH 8) and incubated in 1.25 mM acetylthiocholine and 0.1 mM 5,5′ dithiobis(2-nitrobenzoic acid) at 37 °C, following which the change in solution absorbance at 412 nm was continuously monitored over 1 h using a spectrophotometric plate reader (CLARIOstar®, BMG LabTech, Mornington, VIC, Australia). The concentration of the exosomes was also evaluated using NTA (NanoSight NS300 and NTA 3.2 software, Malvern Panalytical Ltd., Malvern, UK), whereas their size distribution was evaluated from DLS measurements (Zetasizer Nano S, Malvern Instruments Ltd, Malvern, UK) at an emission wavelength of 658 nm.

Additionally, the morphology of the isolated exosomes was visually examined via TEM (1010, JEOL, Frenchs Forest, NSW, Australia) and cryo-EM (Tecnai F30; FEI, Eindhoven, Netherlands). For TEM, 4 μl of isolated exosomes in PBS were adsorbed onto activated Formvar/carbon-coated grids for 10 min followed by incubation in 10 μl 1% uranylacetate for 1 min. The grids were then washed twice in MilliQ® water (18.2 MΩ.cm, Merck Millipore, Bayswater, VIC, Australia) and left overnight to dry. For cryo-EM, a 3-μl aliquot of the purified exosome sample was added onto a holey carbon grid, blotted and plunge-frozen into pre-cooled liquid ethane. Imaging was carried out at an accelerating voltage of 200 kV.

To visualise the MVBs within the cells, cell samples were quickly removed following the experiments and fixed in paraformaldehyde/glutaraldehyde followed by 1% osmium tetroxide and 1.5% potassium ferrocyanide. The fixed cells were then subjected to ethanol dehydration and infiltrated into resin, after which they were sectioned using an ultramicrotome and stained for visualisation under the TEM.

Exosome tracing studies were conducted by tagging the isolated exosomes with BODIPY™ TR ceramide (final dye concentration of 10 μM). Following incubation for 20 min at 37 °C, excess dye was removed using Exosome Spin Columns (MW 3000). The exosomes were then added to unstained recipient cells and incubated for different periods (1, 4 and 18 h), after which the media containing the tagged exosomes was removed. The cells were subsequently washed thrice in PBS and fixed with 4% paraformaldehyde prior to imaging (EVOS M5000, Life Technologies Corp., Bothell, WA, USA). Hoechst 33342 was used as the counterstain.

**Protein and gene profiling.** Exosome marker (ALIX, TSG101, CD63, syntenin-1, flotillin-1, Rab27a and GAPDH) and negative marker (calnexin) proteins were identified from exosomes isolated from both the control and irradiated samples using western blot analysis. Sixty micrograms of the isolated exosomes was lysed in RIPA buffer, which was then denatured in reducing SDS loading buffer (62.5 mM Tris–HCl (pH 6.8), 2% SDS, 25% glycerol, 0.01% bromophenol blue and freshly added 5% $\beta$-mercaptoethanol) by heating at 95 °C for 5 min. The samples were then run on a 10% polyacrylamide gel and transferred onto a nitrocellulose membrane at 60 mV for 60 min. The nitrocellulose membrane was blocked for 1 h

in 5% non-fat skimmed milk in Tris buffered saline solution (TBST; 20 mM Tris, 150 mM sodium chloride, 0.05% Tween® 20) and incubated overnight in the antibody (primary antibody at 1:1000 dilution, anti-mouse antibody at 1:30,000 dilution and anti-rabbit antibody at 1:50,000 dilution) at 4 °C, following which the membranes were treated with the appropriate HRP-conjugated secondary antibody in 0.05% TBST at 37 °C for 1 h. The membranes were then visualised in a gel imager (LI-COR Biotechnology, Lincoln, NE, USA) following incubation in Pierce™ ECL Western blotting detection reagent at room temperature for 2 min. For CD63, non-reducing conditions were employed in which the exosome lysate was mixed with SDS loading buffer but in the absence of $\beta$-mercaptoethanol.

To measure total RNA content, the control and SRBW-treated cells were homogenised using TRiZOL™ and chloroform and centrifuged at $1200 \times g$ to obtain an RNA-containing aqueous layer and a DNA- and protein-containing layer. The RNA was then precipitated with isopropanol and washed in ethanol, dissolved in RNAse-free water with 0.1 µM EDTA and quantified using a UV spectrophotometer (NanoDrop™ One; Thermo Fisher Scientific, Waltham, MA, USA). CDNA was synthesised with the LunaScript® RT SuperMix Kit and RT-qPCR carried out using the Luna® Universal qPCR Master Mix with the aforementioned primers.

**Calcium influx studies**. The cells were seeded at a density of $0.05 \times 10^6$ cells per well in a 24-well glass-bottom plate and incubated overnight in a humidified incubator. After incubation, they were treated with combinations of amiloride HCl (a $Ca^{2+}$ ion channel blocker; 100 µM, for 60 min), thapsigargin (an ER calcium ion inhibitor; 100 nM, for 25 min) and BAPTA-AM (a membrane-permeable calcium chelator that removes intracellular calcium; 10 µM, for 25 min), following which they were incubated in the presence of 10 µM Fura2-AM for 60 min at 37 °C to measure the intracellular $Ca^{2+}$ levels. The cells were then washed to remove the extracellular dye and replenished with DMEM, taking care to protect them from exposure to light. The media devoid of calcium from the inhibitor-treated cells was subsequently replaced with media containing calcium and, where appropriate, exposed to the SRBW irradiation. Changes in the fluorescence intensity were measured with a spectrophotometric plate reader (CLARIOstar®, BMG LabTech, Mornington, VIC, Australia).

**Statistics and reproducibility**. Data presented in this study are expressed as the mean ± the standard error of replicate measurements and analysed using a two-tailed, unpaired Student's $t$ test, where applicable.

**Reporting summary**. Further information on research design is available in the Nature Research Reporting Summary linked to this article.

## Data availability
The data sets generated during and/or analysed during the current study are available from the corresponding author on reasonable request.

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

## Acknowledgements

This work was performed in part at the Materials Characterisation & Fabrication Platform (MCFP) at the University of Melbourne, the Advanced Microscopy Facility at the Bio21 Molecular Science & Biotechnology Institute and the Victorian Node of the Australian National Fabrication Facility (ANFF). L.A.A. is grateful for an Australian Government Research Training Program Scholarship. L.Y.Y. acknowledges support from the Australian Research Council through Discovery Project grant (DP170101061). The authors acknowledge the use of the equipment in the RMIT Microscopy & Microanalysis Research Facility and the RMIT MicroNano Research Facility and are grateful for the technical assistance provided by the staff in these facilities, in particular, Dr. Zeyad Nasa.

## Author contributions

L.Y.Y. and H.L. conceived the original research idea. L.A.A. and S.R. carried out the device development, cell culture, experiments, assays and characterisation with assistance in data analysis from L.Y.Y. L.A.A., S.R., E.H. and C.D. conducted the microscopy. L.A.A., S.R. and L.Y.Y. contributed to, discussed and wrote the manuscript.

## Competing interests

The authors declare no competing interests.
