## [Peer Review File · Communications Biology]

Reviewers' comments:

Reviewer #1 (Remarks to the Author):

In this study, Yeo et al. found that high frequency acoustic stimulation can promote exosome generation by a calcium-dependent manner. Overall, this topic is interesting, especially for exosome-based theranostic fields. However, there are still many unclear questions that have to be solved.

- 1> The description of figure contents in the main text could not match the figures.
- 2> Why the authors chose to use U87-MG cells? If the acoustic stimulation could generate similar effect on other cell lines?
- 3> As shown in figure 1c, the production of exosomes is gradually increased with increasing power. Why the authors chose to use 4W rather than higher values?
- 4> The photo of actual experimental setup needs to be provided for better understanding.
- 5> It is still not clear why the acoustic treatment could result in smaller-sized exosomes (results of figure 2b&2c). Please clarify.
- 6> As shown in figure 1a, the authors utilized silicon oil to couple acoustic device with cells-contained glass well. What is the advantage of silicon oil compared to other coupling agents (such as glue)?
- 7> What is the voltage used in this study?
- 8> As shown in figure 1b, the esterase activity is gradually decreased with prolonging the incubation time from 30 min to 18 hours. It is interesting to see the results of esterase activity after incubating for 10 min and 20 min.

Reviewer #2 (Remarks to the Author):

The authors increased the exosome production around two-fold per hour while keeping the cell viability high using high-frequency acoustic irradiation. They showed that acoustic stimulation increases calcium ion influx and triggers an ESCRT pathway, which is related to exosomal production. This paper demonstrated the possibility of acoustic stimulation to enhance exosome production without adverse effect on cells. The following revisions are required to support claims stated in the manuscript.

1. As the authors pointed out, Yang et al¹ showed 50-fold exosome production increase via nanoporation using nanopores membrane. More descriptions on the unique merit of the reported work should be presented to distinguish the current work from precedent studies with seemingly-better-performance.
2. All the figure numbers were not referenced correctly in the text except Figure 3.
3. Figure 1a shows the chip structure and experimental procedure, which is only explained in the method section. A brief explanation of the principle and workflow would help readers to understand the reported work. Silicon oil and glass-bottomed culture plate are not distinguishable in Figure 1a. Cross-section diagram similar to the author's previous work² will convey the chip structure clearer.
4. In Figure 1b and 1c, more data points are needed between 0 min and 30 min after SRBW irradiation and higher than 4W to determine the optimal incubation time and power for maximum exosome harvest. Additional data points will to explain clearly how control experiments have been conducted at Figure 1 caption.
5. It should be explained why the stimulation duration is 10 minutes while the incubation period is 30 minutes.
6. In the second paragraph of the Result and Discussion, the authors claim that the high-frequency excitation drives reversible membrane permeabilization and responsible for enhancing the secretion of exosomes; however, neither theoretical nor experimental data was provided to support the claim.
7. In figure 2e, data for Calnexin is missing.

8. Based on Figure 3d and Figure 3e, it is not clear why the CD63 level is reached a plateau between n= 4 and n=7.
9. In Figure 4c, the CD63 expression level is comparable to ALIX expression, contradicting the previous result (Figure 3e).
10. In Figure 4, 'Calcium' means the presence or absence of calcium ion in extracellular milieu but missing in the figure caption.

References:

1. Yang, Z. et al. Large-scale generation of functional mRNA-encapsulating exosomes via cellular nanoporation. *Nat. Biomed. Eng.* 4, 69–83 (2020).
2. Ramesan, S., Rezk, A. R., Dekiwadia, C., Cortez-Jugo, C. & Yeo, L. Y. Acoustically-mediated intracellular delivery. *Nanoscale* 10, 13165–13178 (2018).

Reviewer #3 (Remarks to the Author):

In the manuscript entitled "High Frequency Acoustic Cell Stimulation Promotes Exosome Generation Regulated by a Calcium-Dependent Mechanism", Ambattu et al evaluated the acoustic effect on the exosome production in U87-MG cells, and found that acoustic irradiation stimulates the generation of exosome and this process is dependent on the calcium ions. Overall, the finding is novel, and is of general interest. However, the following issues need to be addressed before the acceptance of the manuscript.

1. The authors used the acetylcholine esterase activity to quantify exosome secretion and concluded that the elevated Ach esterase activity suggests the elevated levels of exosomes. No data have shown that Ach esterase activity in single exosome remains the same after their acoustic irradiation treatment. It is possible that exosome number remains the same while Ach esterase activity increases in single exosome. Therefore, the authors need to use other methods to verify their claim.
2. Most of the Figs are wrongly mentioned in the manuscript, and it is hard to follow which Fig the authors are really mentioning. Need to revise it carefully.
3. In Fig 1b, after 30 mins, the authors noticed decreased Ach esterase activity with increasing incubation time and claimed it is possible due to the internalization of the exosomes by neighbouring cells. Any evidence for this claim?
4. With the increase in power, the esterase activity increases almost linearly. No saturation or peak was observed in the numbers they tested. So why the authors choose 4 W for their study?
5. A Cryo-TEM image of exosome and MVB is required to show how the acoustic irradiation may affect the morphology and production of exosomes.
6. In Fig 2d, the authors detected the exosomal protein expression, and found that all the protein involved in exosome production was elevated. However, from their WB result, it is clear that GAPDH, which is not involved in exosome production, is also significantly increase. Why? Meanwhile, the microvesicle and apoptotic body markers need to be checked to verify no contamination of other EVs in the sample.
7. In Fig 4a and 4d, the results showed the depletion of calcium in the extracellular + SRBW completely blocks the SRBW effect on exosome production. This is in contradict with current knowledge that SRBW may increase the intracellular calcium concentration from the intracellular Ca store. Any explanation on this?

8. The authors need to verify their findings in other cell lines.

Specific Responses to Reviewer #1

We thank the reviewer for their careful consideration of our work and are grateful that they have found the topic interesting. We have attempted to address each of the reviewer's comments in a revision and hope that this has led to a stronger manuscript.

1> The description of figure contents in the main text could not match the figures.

We do sincerely apologise for confusing the reviewer due to an error in cross-referencing the figures in the original manuscript. We have corrected for this in the revised manuscript.

2> Why the authors chose to use U87-MG cells? If the acoustic stimulation could generate similar effect on other cell lines?

The acoustic stimulation in our study provides a facile and economical platform to stimulate the same cells repeatedly without adversely affecting their viability. We used a U87-MG cell line which represents a less accessible glioblastoma so that heterogeneity of the exosome population can be reduced. However, as suggested by the reviewer, we have now repeated the studies to demonstrate the applicability of the method to another cell line (A549).

3> As shown in figure 1c, the production of exosomes is gradually increased with increasing power. Why the authors chose to use 4W rather than higher values?

With increasing power beyond 4 W, the cell viability dropped from 80-70% (we have now included additional data to show this in Supplementary Figure S3a). Higher powers were therefore avoided since the exosomes produced can reflect the stressed cells. We also wanted to avoid contamination with apoptotic bodies and apoptotic exosomes which might be formed under reduced cell viability conditions. A brief explanation has now been added to the revised manuscript on page 5.

4> The photo of actual experimental setup needs to be provided for better understanding.

We have now added an image of the experimental setup as S1a in supplementary information.

5> It is still not clear why the acoustic treatment could result in smaller-sized exosomes (results of figure 2b&2c). Please clarify.

The acoustic stimulation produced a heterogenous population of exosomes with sizes ranging from 30 nm to 150 nm, as shown in Figure 2b. As the reviewer pointed out, an increase in smaller vesicles can be seen with the acoustically stimulated cells. This may be due to a difference in the proteomic profile of the stimulated cells. The acoustically stimulated cells had significantly higher ALIX expression compared to that of CD63. Prior reports (e.g., Colombo et al., *J Cell Sci* 126, 5553, 2013) have shown that ALIX determines the heterogeneity of the exosome population depending on the cell type although the exact mechanism which guides this is still unclear. Gruber et al. (*Mol. Ther.* 26, 634, 2018) has also shown increase in ALIX and TSG101 expression associated with exosomes with sizes ranging from 100 – 200 nm.

From these, we speculate that the increase in ALIX expression along with that for TSG101 could have favoured the production of exosomes with smaller dimensions.

6> As shown in figure 1a, the authors utilized silicon oil to couple acoustic device with cells-contained glass well. What is the advantage of silicon oil compared to other coupling agents (such as glue)?

The acoustic impedance for silicon oil is relatively low (Zhang et al., Appl Phys Lett 102, 163702, 2013) compared to other materials such as glue. Moreover, silicon oil is non-toxic and has been used in various medical applications. Additionally, the viscosity of the silicon oil allowed us to deposit it onto the acoustic device uniformly whilst allowing ease of removal without damaging the device which would be hard to achieve with PDMS or glue.

7> What is the voltage used in this study?

The optimised voltage used in the study corresponding to a power of approximately 4 W is 100 Vpp. 50 Vpp and 80 Vpp correspond to powers of 1 W and 2.5 W, respectively.

8> As shown in figure 1b, the esterase activity is gradually decreased with prolonging the incubation time from 30 min to 18 hours. It is interesting to see the results of esterase activity after incubating for 10 min and 20 min.

As suggested by the reviewer, the esterase activity of exosomes collected after 10 and 20 minutes of post exposure incubation has now been included in Fig. 1b in the revised manuscript.

Specific Responses to Reviewer #2

The authors increased the exosome production around two-fold per hour while keeping the cell viability high using high-frequency acoustic irradiation. They showed that acoustic stimulation increases calcium ion influx and triggers an ESCRT pathway, which is related to exosomal production. This paper demonstrated the possibility of acoustic stimulation to enhance exosome production without adverse effect on cells. The following revisions are required to support claims stated in the manuscript.

We thank the reviewer for their careful consideration of our work. We have attempted to address each of the reviewer's comments in a revision and hope that this has led to a stronger manuscript.

1. As the authors pointed out, Yang et al¹ showed 50-fold exosome production increase via nanoporation using nanopores membrane. More descriptions on the unique merit of the reported work should be presented to distinguish the current work from precedent studies with seemingly-better-performance.

The limitations of precedent techniques was discussed in the original work (page 4 in the revised manuscript). The nanoporation technique results in an increase in intracellular calcium and also heat shock proteins (HSP), the latter being activated when the temperature rises 5 degrees above the ambient temperature. Such an elevation of HSP was observed to result in p53-directed TSAP-6 induced exosome production. However, such exosomes have also been reported to induce cell death, pro-invasive niches or to produce tumor supporting macrophages (Cooks et al., Nat Commun 9, 771, 2018; Novo et al., Nat Commun 9, 5069, 2018). Moreover, it is not clear that the nanoporation technique is suitable for production of exosomes from the same cells through repeated treatment as we have shown possible with our technique in which we show the possibility for maintaining exosome population homogeneity. This is now briefly discussed on page 4 of the revised manuscript. The advantage of maintaining exosome population homogeneity, on the other hand, has been discussed throughout the manuscript, for example, in the abstract, on pages 4-5 and 7-8 of the revised manuscript, and in the conclusion.

2. All the figure numbers were not referenced correctly in the text except Figure 3.

We do sincerely apologise for confusing the reviewer due to an error in cross-referencing the figures in the original manuscript. We have corrected for this in the revised manuscript.

3. Figure 1a shows the chip structure and experimental procedure, which is only explained in the method section. A brief explanation of the principle and workflow would help readers to understand the reported work. Silicon oil and glass-bottomed culture plate are not distinguishable in Figure 1a. Cross-section diagram similar to the author's previous work² will convey the chip structure clearer.

We have now revised the figure according to the reviewer's suggestions.

4. In Figure 1b and 1c, more data points are needed between 0 min and 30 min after SRBW irradiation and higher than 4W to determine the optimal incubation time and power for maximum exosome harvest. Additional data points will to explain clearly how control experiments have been conducted at Figure 1 caption.

Additional data points have now been included in Figure 1b and Supplementary Figure S3a as recommended by the reviewer. Briefly, higher powers beyond an optimum of around 4 W led to a reduction in the cell viability. Since the exosomal content depends on the condition of the parent cell, we limited the powers to those values that led to improved exosome production but without affecting the cell viabilities. A brief explanation has now been added to the revised manuscript on page 5.

5. It should be explained why the stimulation duration is 10 minutes while the incubation period is 30 minutes.

The stimulation and incubation times were estimated from the optimisation studies. Acoustic exposure over 10 minutes gave better esterase activity while exposure beyond that duration was found to be detrimental to the cells. The data to show this optimum point is now included in the Supplementary Fig. S3.

6. In the second paragraph of the Result and Discussion, the authors claim that the high-frequency excitation drives reversible membrane permeabilization and responsible for enhancing the secretion of exosomes; however, neither theoretical nor experimental data was provided to support the claim.

The possibility for reversibility in the membrane permeabilization was experimentally shown by Reusch et al., Phys Rev Lett 113, 118102, 2014 and Ramesan et al, Nanoscale 10, 13165, 2018 (Ref. 55 in the revised manuscript). The former study, which we have added as a reference (Ref. 57) to the revised manuscript, also included a theoretical description for this.

7. In figure 2e, data for Calnexin is missing.

The calnexin content was not detectable in the Western blot and hence its relative band intensity was not included. We have nevertheless included this now in the revised figure.

8. Based on Figure 3d and Figure 3e, it is not clear why the CD63 level is reached a plateau between n= 4 and n=7.

The heterogeneity in the exosome markers is still unclear and there is a huge gap in understanding the mechanism by which exosome cargoes are determined from their parent cells. Having said that, a number of arguments could be made to speculate why the CD63 expression is plateauing beyond a point.

Both CD63 and ALIX take part in endosomal trafficking and the fate of these vesicles are difficult to determine. However, studies have shown that apoptotic or stressed cells have a tendency to remove CD63 into the extracellular space through various mechanisms, one of which being exosome secretion. There are reports suggesting that CD63 expression is

elevated in apoptotic exosomes rather than regular exosomes while ALIX expression is significant only in regular exosomes (Park et al., PNAS 115, E11721, 2018). Moreover, overexpression of TSG101 along with ALIX avoids apoptotic predisposition of ALIX in stressed cells (Kaul & Chakrabarti, Mol Biol Cell 28, 2106, 2017). Since the acoustic treatment used in the study is a cell-friendly technique that preserves the cell viability, it may be concluded that the acoustic treatment leads to secretion of regular exosomes as opposed to apoptotic exosomes, and this may be a reason why the plateau in the CD63 expression was observed.

Additionally, it is known that the affinity of syntenin-1 for CD63 is 10-fold lower than its affinity for syndecans, which mediate the interaction between ALIX and syntenin (Baietti et al., Nat Cell Biol 14, 677, 2012). As such, it can be concluded that ALIX is favoured over CD63 in the biogenesis of regular exosomes, which might also explain the plateau in the CD63 expression.

9. In Figure 4c, the CD63 expression level is comparable to ALIX expression, contradicting the previous result (Figure 3e).

ALIX (ALG-2 binding protein X) is a calcium dependent protein (Missotten et al., *Cell Death Differ* 6, 124, 1999; Sun et al., *Cell Discov* 1, 15018, 2015). Hence in the presence of calcium, its expression is expected to be affected (Figs. 3e and 3f). In other words, as the calcium content increases, we expect the ALIX expression to also change. CD63, on the other hand, has no known association with calcium. In Figs. 4c and 4d, the calcium availability is reduced in the presence of the inhibitor and chelator treatments. As such, the ALIX expression can be seen to become comparable to that of CD63. We have added a sentence as a footnote on page 11 to explain this.

10. In Figure 4, 'Calcium' means the presence or absence of calcium ion in extracellular milieu but missing in the figure caption.

We thank the reviewer for pointing this out and have corrected for this in the figure.

Specific Responses to Reviewer #3

In the manuscript entitled “High Frequency Acoustic Cell Stimulation Promotes Exosome Generation Regulated by a Calcium-Dependent Mechanism”, Ambattu et al evaluated the acoustic effect on the exosome production in U87-MG cells, and found that acoustic irradiation stimulates the generation of exosome and this process is dependent on the calcium ions. Overall, the finding is novel, and is of general interest. However, the following issues need to be addressed before the acceptance of the manuscript.

We thank the reviewer for their careful consideration of our work and are grateful that they have found the work novel and of interest. We have attempted to address each of the reviewer’s comments in a revision and hope that this has led to a stronger manuscript.

1. The authors used the acetylcholine esterase activity to quantify exosome secretion and concluded that the elevated Ach esterase activity suggests the elevated levels of exosomes. No data have shown that Ach esterase activity in single exosome remains the same after their acoustic irradiation treatment. It is possible that exosome number remains the same while Ach esterase activity increases in single exosome. Therefore, the authors need to use other methods to verify their claim.

Acetylcholine esterase is a known method to assess the quality and quantity of exosomes. As the reviewer pointed out, the increase in esterase activity can however be a function of increased exosomal cargo, which is why we quantified the number of exosomes independently via nanoparticle tracking analysis (Nanosight). This result is reported in Figure 2a in which an increase in the number of exosomes can be clearly seen following acoustic stimulation. We have also included two videos from the nanoparticle tracking analysis that show significantly more exosomes in the SRBW irradiated sample compared to that in the unirradiated control.

2. Most of the Figs are wrongly mentioned in the manuscript, and it is hard to follow which Fig the authors are really mentioning. Need to revise it carefully.

We do sincerely apologise for confusing the reviewer due to an error in cross-referencing the figures in the original manuscript. We have corrected for this in the revised manuscript.

3. In Fig 1b, after 30 mins, the authors noticed decreased Ach esterase activity with increasing incubation time and claimed it is possible due to the internalization of the exosomes by neighbouring cells. Any evidence for this claim?

The role of exosomes as messengers in cell–cell communication and their internalisation by neighbouring cells has been reported in various studies (see, for example, Fevrier et al., PNAS 101, 9683, 2004 and Delencos et al, Front Neurosci 11, 172, 2017). However, to show this in our work, the exosomes isolated from the cells were tagged and then incubated with untagged cells. The internalisation becomes visible after 4 hours of incubation and is significantly visible after 18 hours. This data is now included in Fig. S2 in the Supplementary Information.

4. With the increase in power, the esterase activity increases almost linearly. No saturation or peak was observed in the numbers they tested. So why the authors choose 4 W for their study?

As the power is increased beyond approximately 4 W, the cell viability was observed to decrease (we have now included additional data to show this in Supplementary Figure S3a). As the stress to the cell can alter the exosomal cargo, we limited ourselves to this power to maximise exosome production whilst minimising adverse effects to cell viability. A brief explanation has now been added to the revised manuscript on page 5.

5. A Cryo-TEM image of exosome and MVB is required to show how the acoustic irradiation may affect the morphology and production of exosomes.

As suggested by the reviewer, a cryo-TEM of the exosomes and MVBs have now been included in Fig. 2c and Supplementary Figure S5 in the revised manuscript.

6. In Fig 2d, the authors detected the exosomal protein expression, and found that all the protein involved in exosome production was elevated. However, from their WB result, it is clear that GAPDH, which is not involved in exosome production, is also significantly increase. Why? Meanwhile, the microvesicle and apoptotic body markers need to be checked to verify no contamination of other EVs in the sample.

GAPDH is a house keeping gene and hence GAPDH protein is found in the exosomes and, in fact, is the third most common protein in exosomes according to the exosome database ExoCarta. This has also been reported in the literature (see, for example, Stamer et al., J Proteomics 74, 796, 2011). Although GAPDH is not a marker exclusive for exosomes, it is an essential protein found in exosomal proteome. As such, it would not be a surprise that the GAPDH content increases with an increase in the exosomes.

Calnexin acts as a positive marker for both apoptotic bodies and microvesicles (Guérin et al. Mol Biol Cell 19, 4404, 2008; Haraszti et al., J Extracell Vesicles 5, 32570, 2016; Willms et al., Sci Rep 6, 22519, 2016). Calnexin was not detected in the exosomes that were isolated in our work, which therefore provides the verification the reviewer requested to show that the presence of apoptotic bodies and microvesicles were not significant.

7. In Fig 4a and 4d, the results showed the depletion of calcium in the extracellular + SRBW completely blocks the SRBW effect on exosome production. This is in contradict with current knowledge that SRBW may increase the intracellular calcium concentration from the intracellular Ca store. Any explanation on this?

We believe that the reviewer may have misunderstood our statement on this in the original manuscript which may have led him/her to think there is a contradiction. In general, any stress to the cell may cause an increase in intracellular calcium either through release from intracellular calcium store or by internalizing calcium from the extracellular milieu. In our specific case of the SRBW stimulation though, we only observed an increase in the intracellular calcium level through the latter (i.e., uptake from the external milieu) rather than the former (i.e., release from the intracellular calcium store).

To avoid the ambiguity which we believe has possibly led to the reviewer's misunderstanding, we have reworded the text and relabelled 'calcium' as 'extracellular calcium' in the table in Figure 4. We have also added an additional dataset (labelled as iii) to show that the acoustic stimulation alters the intracellular calcium level only through uptake of calcium from the extracellular milieu.

8. The authors need to verify their findings in other cell lines.

As suggested by the reviewer and as recommended by the editor, results showing the effect of acoustic stimulation on an additional cell line (A549) has now been included.

REVIEWERS' COMMENTS:

Reviewer #1 (Remarks to the Author):

All my questions have been solved properly, I can recommend it to be published as it is.

Reviewer #2 (Remarks to the Author):

The authors addressed the raised concerns and revised the manuscript accordingly. The manuscript can be published with minor corrections listed below.

1. What is the justification of choosing immortalized glioblastoma and lung cancer cells if the ultimate goal of this technology is to promote the utilization of exosomes for therapeutic purposes? Are these cells commonly used for such purposes? What would be the anticipated performance if more fragile cells (primary cells or stem cells) are acoustically stimulated?
2. Need a space between ref 44 and "Given" on page 4.
3. Fig 2d Sytenin-1 band images for Control and SRBW exhibit an artificial contrast line as if one of the images was cut and pasted (not from the gel as other images in the panel are).
4. Footnote 1 on page 11 seems odd to have for a scientific manuscript.

Reviewer #3 (Remarks to the Author):

I feel that the authors have adequately conducted additional experiments and provided further data to support their findings which I had questions about. Therefore, I have no further comments to add pertaining to the revised manuscript.

Specific Responses to Reviewer #2

The authors addressed the raised concerns and revised the manuscript accordingly. The manuscript can be published with minor corrections listed below.

We thank the reviewer for kindly and carefully assessing our revised manuscript, and are pleased to learn that we have addressed most of their concerns.

1. What is the justification of choosing immortalized glioblastoma and lung cancer cells if the ultimate goal of this technology is to promote the utilization of exosomes for therapeutic purposes? Are these cells commonly used for such purposes? What would be the anticipated performance if more fragile cells (primary cells or stem cells) are acoustically stimulated?

We first note that the ultimate aim of the technology is to enhance exosome production not just for therapeutic but also for diagnostic purposes. Having more exosomes to sample in the latter application would lead to better detection sensitivity, particularly when there are limited physiological sample quantities available. We have chosen to demonstrate proof-of-concept in cancer cells as they are commonly used in exosome release studies and since exosomes derived from cancer cells have been identified as biomarkers for early cancer detection (Wee et al., *Biochim Biophys Acta Rev Cancer* 1871, 12, 2019) or as vaccines for cancer immunotherapy (Bae et al., *Genes Cancer* 9, 87, 2018). We have added a note on page 4 in the revised text to highlight this.

In preliminary studies, we have also observed enhancement of exosome release when we acoustically stimulate human mesenchymal stem cells. Their post-excitation viability however is dependent on their passage number and the number of excitation–incubation cycles. We are currently studying this in further detail, and intend to report this in a subsequent publication.

2. Need a space between ref 44 and “Given” on page 4.

We did not observe this in our version but would be happy to address it in the proof if necessary.

3. Fig 2d Sytenin-1 band images for Control and SRBW exhibit an artificial contrast line as if one of the images was cut and pasted (not from the gel as other images in the panel are).

We sincerely apologise for not clearly differentiating the spliced image. In the revised manuscript we have now added a demarcating line between the two blots and have provided the entire scan of the syntenin-1 blot in Supplementary Figure 5.

4. Footnote 1 on page 11 seems odd to have for a scientific manuscript.

We have now removed the footnote and added the text to the manuscript.